# Genetic and Epigenetic Etiology Underlying Autism Spectrum Disorder

**DOI:** 10.3390/jcm9040966

**Published:** 2020-03-31

**Authors:** Sang Hoon Yoon, Joonhyuk Choi, Won Ji Lee, Jeong Tae Do

**Affiliations:** Department of Stem Cell and Regenerative Biotechnology, KU Institute of Technology, Konkuk University, Seoul 05029, Korea; kei99137@naver.com (S.H.Y.); uccoschoi1@gmail.com (J.C.); dnjs303@naver.com (W.J.L.)

**Keywords:** autism spectrum disorder, genetic, epigenetic, etiology

## Abstract

Autism spectrum disorder (ASD) is a pervasive neurodevelopmental disorder characterized by difficulties in social interaction, language development delays, repeated body movements, and markedly deteriorated activities and interests. Environmental factors, such as viral infection, parental age, and zinc deficiency, can be plausible contributors to ASD susceptibility. As ASD is highly heritable, genetic risk factors involved in neurodevelopment, neural communication, and social interaction provide important clues in explaining the etiology of ASD. Accumulated evidence also shows an important role of epigenetic factors, such as DNA methylation, histone modification, and noncoding RNA, in ASD etiology. In this review, we compiled the research published to date and described the genetic and epigenetic epidemiology together with environmental risk factors underlying the etiology of the different phenotypes of ASD.

## 1. Introduction

Autistic disorder or a broader form of autism spectrum disorder (ASD) is a neurodevelopmental disorder characterized by difficulties in social interaction, delayed development of communication and language, repeated body movements, and impaired intelligence development, first described by psychiatrist Leo Kanner in 1943 [1,2]. The prevalence of typical autism and ASD is approximately 5.5–20 and 18.7–60 per 10,000 individuals, respectively [3]. Moreover, ASD has increased steadily since the term was coined, with the prevalence of autism worldwide currently at 1%–2% [4,5,6,7]. This phenomenon is partly due to an increase in awareness and the development of the Mental Disorders Diagnosis and Statistical Manual (DSM) criteria, starting with schizophrenia which began in 1952, and the development of key diagnostics, which currently deal with various mental disorders [8]. In addition, about 31% of ASD patients showed intellectual disabilities [9] and 20%–37% of them were known to have epilepsy [10,11]. Moreover, ASD is often accompanied by psychiatric or other medical problems, including anxiety disorders, depression, attention deficit hyperactivity disorder, sleep disorders, and gastrointestinal problems [12,13,14]. So far, many theories about ASD etiology and pathogenesis have been proposed, but it is said to be related to the interaction of genetic and environmental factors [15,16]. The concordance rate of ASDs in monozygotic twins (92%) was much higher than that in dizygotic twins (10%), indicating that genetic factors are more likely to contribute to ASD than environmental factors [17]. Genome-wide association and microscopy analysis have identified many different loci and genes that are associated with the etiology of ASD. However, although many genetic and epigenetic risk factors have been suggested, no clear pathogenesis and specific diagnostic markers for ASD have been identified. Accumulated evidence has also demonstrated an important role of epigenetic factors, such as DNA methylation, in ASD etiology [18]. To better understand the molecular basis of ASD, we describe the genetic and epigenetic epidemiology along with the environmental risk factors underlying the etiology of ASD (Figure 1). 

## 2. Environmental and Prenatal Factors that Cause ASD

### 2.1. Viral Infection

As ASD is a neurodevelopmental disorder, it is commonly considered a genetic disorder. However, there are many studies that support the idea that environmental factors can be a major cause of ASD. Most of these factors are due to the prenatal period, which can be affected by environmental changes within the parental body [19]. During pregnancy, the maternal body becomes immunosuppressed, which makes the mother and the developing embryo susceptible to many infectious agents [20]. Similarly, it has been consistently suggested that parental viral infections are associated with the development of autism in their offspring [21,22,23]. Among the infectious diseases, some have been specifically pointed out as contributing to infantile autism when infection occurs during the first trimester of pregnancy [24]. These diseases include rubella [21,25,26,27], measles, mumps [21,26,27], chicken pox [21,28], influenza [21,23], herpes simplex virus [29], pneumonia, syphilis, varicella zoster [30], and cytomegalovirus [27,31,32,33]. Of note, cytomegalovirus is known to cause permanent neurological damage in about 10%–20% newborns when the mother is infected [34]. Moreover, bacterial infection during the second trimester of pregnancy has also been suggested to cause autism of infants [21,22]. In some cases, autoimmune diseases of parents were shown to be related to infantile autism [35,36]. Animal model studies have also shown that maternal infections activated the immune system, which eventually affects fetal brain development [37,38].

### 2.2. Parental Age

We have shown that maternal infection is directly related to the fetal pathological status since the baby is developed and nourished within the maternal body [20]. Similarly, the age of pregnant women and paternity were suggested as one of the most plausible contributors to increasing the risk of autism [39,40,41,42,43,44,45]. Meta-analysis for the correlation between maternal age and autism was analyzed by Sandin et al. A maternal age < 20 showed a lower risk (the relative risk for autism was 0.76) for autism compared to a maternal age between 25 and 29. On the other hand, the relative risk for mothers aged 35 or over compared to mothers aged 25 to 29 was 1.52 [46]. Reichenberg et al. reported a population-based study showing that the risk for autism began to increase at the paternal age of 30 and continued to increase after the age of 50 [42]. Paternal ages above 55 had at least twice the risk to have a child with autism, compared with those below 50 [42]. Moreover, it is well known that the older the parents, the higher the chance of miscarriage [47,48,49,50], fetal death [47,51,52], childhood cancers [53,54], and schizophrenia [55,56,57,58]. This is thought to be due to an increase in de novo genetic mutation during germ cell development in the aging process [59,60]. The effect of parental age on various diseases has been supported by many studies, and the correlation between parental age and autism seems to be one of the most acceptable factors causing autism [47,53,55,59].

### 2.3. Zinc Deficiency

The physiological function of zinc was first identified in the study of carbonic anhydrase [61]. Currently, more than 300 zinc-related enzymes have been discovered, including isoenzymes [62]. Zinc, as a cofactor in metalloenzymes, plays a catalytic role mostly in the transformation of substrates by aiding the formation of hydroxide ions at neutral pH or through Lewis acid catalysis [63,64]. Zinc is now known to be an essential trace element that plays a role in the immune system, protein synthesis, and wound healing [65]. Moreover, zinc has been known to play a role in forming the zinc finger motif of proteins and binding them to DNA, suggesting that zinc is also involved in regulating gene expression [66]. Zinc also supports fetal growth and development during pregnancy and the development of children [67,68]. Therefore, prolonged deficiency of zinc during pregnancy might lead to diverse dysfunction of embryonic growth, especially neurodevelopment [69,70,71]. Research on the relationship between zinc and autism began with reports of the metal ion’s involvement in neurodegeneration and dysfunction [72,73,74,75,76,77]. Since metal toxicity was shown to cause damage to the central nervous system [71,72], it was expected that an excess of zinc could cause damage to the nervous system [78,79,80,81,82,83,84,85]. A recent study also suggests that a toxic metal uptake and deficiency of essential elements increase the risk of ASD [86]. It has been noted that zinc interacts with β-amyloid and its precursors, which are crucial factors for the degenerative process of the brain [73,85,87,88,89,90,91]. 

Synaptic morphology and function were associated with autism, schizophrenia, and Alzheimer’s disease. The normal function of synapses depends largely on the molecular setting of the synaptic proteins, including ProSAP/Shank proteins, which function as scaffolding molecules for protein–protein interaction at postsynaptic density. ProSAP/Shank localization to postsynaptic density is induced by increased levels of zinc [92,93,94]. Thus, zinc deficiency was shown to dysregulate ProSAP/Shank and postsynaptic density in vivo and in vitro [94]. Several reports suggested that mutation in ProSAP/Shank could lead to ASD [95,96,97]. Moreover, ProSAP/Shank proteins, including ProSAP1/Shank2 and ProSAP2/Shank3, have a C-terminal sterile alpha motif, to which zinc can bind [98]. Thus, a lack of zinc prevents the zinc-dependent ProSAP/Shank proteins from playing a normal role in the formation of the scaffold structure. This leads to synaptic defects and can also lead to autism [94,99]. Moreover, the relation of zinc uptake and the expression of *Shank3* regarding autism has been studied recently [100]. This study only included participants with genetically confirmed diagnosis of Phelan McDermid Syndrome (PMDS) with deletion of *Shank3* gene [100]. The study showed that low Shank3 levels resulted in abnormally low zinc transporter, which led to low zinc concentration [100]. Statistical data also suggests the close relationship between zinc deficiency and infantile autism [101]. Of children between 0 and 3 years of age with autism, 43.5% (251/577) were zinc deficient in males and 52.5% (62/118) in females [101]. Among autism children from 4 to 9 years old, high rates of zinc deficiency were still found in males (28.1%) and females (28.7%) [101]. An animal-based study of Shank3^+/-^ and Shank^-/-^ transgenic mouse compared with prenatal zinc-deficient autism mouse model, which are offspring from zinc-deficient diet fed mice, showed diverse brain region abnormalities in different models of ASD [102]. However, the role of zinc deficiency on autism is still controversial. Sweetman et al. tested blood sample of 74 ASD children and claimed that zinc deficiency was not related to ASD [103,104]. Another recent report also suggested that zinc deficiency may not be micronutrient deficiency during pregnancy but may be a compensatory mechanism to prevent exposure to air pollutants during fetal development [103,104].

## 3. Genetic Epidemiology

Because autism is a neurodevelopmental disorder, the genetic aspect of autism has continuously been studied along with other factors that affect neurodevelopment. Autism can be defined by three behavioral domains: social interaction, language communication and imaginative play, and range of interests and activities [105,106]. Therefore, studies have been conducted to find genes involved in each symptom of ASD and to identify how the genes are related to ASD pathogenesis. Due to various symptoms of ASD, ASD-related genes are also closely related to other neurodevelopmental syndromic disorders, such as fragile X syndrome and Rett syndrome [107]. 

### 3.1. Chromosome Loci that Affect ASD

The phenotype of ASD is very vague and the specific factors that cause the disease have not yet been clearly elucidated [108]. The diverse phenotypes of ASD may be due to the large number of genes or environmental factors involved in autism, resulting in different genetic variations that occur in each individual. Moreover, the diverse interaction between the various ASD-associated genes makes it difficult to interpret its pathogenesis [109]. Previous research has approached this uncertainty by identifying gene mutation or copy number variation (CNV) at specific chromosomal loci that are relevant to neurodevelopment among individuals and families [109,110,111,112]. Genes that have long been mentioned as involved in the causation of autism are *FOXP2, RAY1/ST7, IMMP2L*, and *RELN* genes at 7q22-q33 [109]. These genes are also involved in diagnosable diseases that are associated with autism such as neurofibromatosis, tuberous sclerosis complex, and fragile X syndrome [109]. Moreover, not only coding regions but also noncoding regions of risk genes were found to be related to the etiology of autism [113].

### 3.2. Candidate Genes on Chromosome 7 

Some researchers have focused on the locus of chromosome 7, as many genes in this region seem to be related to autism [114,115,116] (Figure 2). An international study conducted by the International Molecular Genetic Study of Autism Consortium on 99 multiplex families was able to point out the most probable ASD-related regions on six different chromosomes (chromosomes 4, 7, 10, 16, 19, and 22), with chromosome 7 being the most significant [117]. Interestingly, a genome-wide study on the pedigree of the KE family have shown that the region implicated in language disabilities is localized to 7q31 [118]. Half of the KE family members struggled with serious language impairment. Genome-wide association studies found more specific loci concerning speech and language development and autism-related loci, such as SPCH1 and AUTS1 (autism susceptibility locus) [119,120]. Subsequent studies also reported increased allele sharing on 7q within the autism relative pair families [115,121,122]. 

#### 3.2.1. FOXP2

*Forkhead box P 2 (FOXP2)* is the first known gene involved in oral movement and speech [123,124]. Heterozygous *FOXP2* mutation causes severe speech and language disorders, while cognition and other aspects are relatively low in severity. *FOXP2* is localized between 7q31.1 and 7q31.31, which are known to be associated with language impairment and mental retardation. Further research discovered that *FOXP2*, as a transcription factor, could regulate gene expression in the development of lung, cardiovascular, intestinal, and neural tissues [124,125]. The inheritance pattern of the three-generation pedigree of the KE family revealed that all affected family members had a point mutation in the forkhead domain of *FOXP2*, indicating a relevance of this gene to the speech and language deficits observed in ASD [123,126]. However, there are also contradictory views on the relationship between *FOXP2* and ASD [127,128]. Case-control association study with Spanish ASD patients revealed that common variants of FOXP2 were not directly connected with ASD [127,129].

#### 3.2.2. RAY1/ST7

The *RAY1/ST7* gene was identified in an autistic individual carrying a translocation breakpoint on chromosome 7 [113]. *RAY1*, which spans more than 220 kb of DNA and is encoded by 16 exons by alternative splicing, is expressed in a variety of tissues with varying levels of expression [113]. However, any specific variants in the coding region were not found in 27 unrelated ASD individuals, indicating that there is no sequence associated with etiology of ASD in the coding sequence [113]. It was also suggested that long noncoding RNA (lnc RNA) called ST7 overlapping transcript antisense 1-4 (ST7OT1-4), which possibly regulates the expression of *RAY1/ST7*, could be associated with autism [130,131]. Moreover, no significant further studies have been conducted to provide solid evidence for the relationship between *RAY1/ST7* and ASD since the study of Vincent et al. [131]. Therefore, further studies have to be made to verify the function of *RAY1/ST7* and its implications in autism [131].

#### 3.2.3. IMMP2L

*IMP2 inner mitochondrial membrane protease-like* (*IMMP2L*) was identified as the most frequently associated gene to autism through high-density SNP (single nucleotide polymorphism) analysis on chromosome 7 [132]. *IMMP2L* was originally known to be related to a complex neuropsychiatric disorder, Gilles de la Tourette syndrome, which demonstrated an overlapping phenotype with ASD [133]. A high-density association analysis study was conducted with 127 families and 188 gender-matched controls that focused on the locus *autism, susceptibility to, 1* (*AUTS1*) of chromosome 7 [132]. This study screened more than 3000 SNPs in the conserved region and highlighted several genes including *IMMP2L* and *dedicator of cytokinesis 4* (*DOCK4*) that required further research to determine their association with ASD [132]. Fabian et al. recently conducted an animal-based study using *IMMP2L* knockdown mice to study the effect of *IMMP2L* deficiency on behavioral domains. This study shows that IMMP2L deficiency induced behavioral effects, which was gene-dose and sex dependent [134]. However, *IMMP2L* may be not a common gene that causes ASD because coding mutation was not observed in ASD patients [135]. Another family-based association study focused on *zinc finger protein 533* (*ZNF533*), *DOCK4*, and *IMMP2L* genes in the Chinese Han population showed that SNPs within *ZNF533* and *DOCK4* were related to autism, whereas *IMMP2L* was shown irrelevant [136].

#### 3.2.4. RELN

*Reelin* (*RELN*) is a gene with 65 exons located in 7q22 that is necessary for the formation of brain structure by directing the migration of several neuronal cell types and the development of neural connections [137,138,139,140]. The signaling protein roles of *RELN* in the migration of neurons and neural connection could explain the *RELN* abnormalities in patients with ASD [141,142,143,144,145] along with Alzheimer’s disease [146,147], schizophrenia [148,149], lissencephaly [150,151], and bipolar disease [149]. One study showed that reduced blood levels of *RELN* might be the cause of autism [152]. In addition, to determine the association between ASD and single-locus markers and multi-locus haplotypes, family-based association analysis for 218 Caucasian families showed *RELN* as an important potential contributor to autism [143]. Moreover, a larger family-based RNA-SSCP and DNA sequencing data revealed the association and linkage that a polymorphic trinucleotide repeat (GGC) located in the 5′ untranslated region of the *RELN* gene may play a role in the transcriptional regulation, and longer GGC repeats are correlated with vulnerability to ASD [142]. Wang et al. reviewed and analyzed papers published in 2013 and concluded that rs362691 SNP in *RELN* contributed more to the ASD risk than rs736707 or GGC repeat variants [153].

According to the Simons Foundation Autism Research Initiative (SFARI) database [154], 913 genes and 17 recurrent CNV loci are suggested to be implicated with autism. Among these 17 CNV loci, only one CNV locus, 7q11.23, is known to exist within 7q22-33 region [155]. Moreover, based on the gene score database conducted by SFARI, *RELN* is the only gene which seems to have a strong association to autism on 7q [156,157]. Therefore, further studies are required to narrow down the potential genes that are associated with ASD.

### 3.3. Neurodevelopmental Disorders and ASD by CNV

In addition, recent studies have found that inherited and de novo CNV could contribute to ASD. Changes in genetic expression involved in neural development are the main genetic etiology of ASD, such as suppression of neurodevelopment, changes in brain size, synapse formation, and connectivity between brain regions. These gene dosage changes can be caused by CNV and can be confirmed by SNP analysis. CNVs are a phenomenon in which parts of the genome are repeated in various numbers from individual to individual by deletion, duplication, translocation, and reversal [158]. Seven percent of ASD families were found to be associated with de novo CNVs. For example, duplications in 16p11.2, 15q11-q13, 7q11.23, 1q21.1, 22q11.2, and 7q22-q31 and deletions in 16p11.2, 3q29, and 22q11.2 were found to be associated with ASD [159] (Table 1). Microdeletion of 16q24.3 is associated with ASD because it affects *ankyrin repeat domain 11* (*ANKRD11)* and *zinc finger protein 778* (*ZNF778)* genes, leading to cognitive impairment and brain abnormality [160]. Chromosomal deletion of ~593 kb or segmental duplication of ~147 kb in 16p11.2 affected the neural development of the brain, which caused ASD [161,162]. These 16p11.2 microdeletions and microduplications were found in approximately 1% of patients with ASD. *Potassium channel tetramerization domain-containing 13* (*KCTD13*), which is one of the genes that encompasses the deletion regions in 16p11.2, is associated with neurodevelopmental phenotypes. *KCTD13* encodes a polymerase delta-interacting protein 1 that interacts with polymerase δ in the nucleus of proliferating cells. Therefore, *KCTD13* deletion resulted in a decrease in the proliferation of neuronal progenitors and an increase in cell death during brain development [163]. On the other hand, overexpression of *KCTD13* with an increase in redundancy of 16p11.2 delayed brain development and caused microcephaly. The effect of *KCTD13* suppression is still controversial. Golzio et al. argued that the deletion of 16p11.2 stimulated brain development, which evolved into macrocephaly [164]. This was rebutted by Escamilla et al., who showed no differences in embryonic nor adult brain size after *KCTD13* deletion in mice [165]. 

The 5p14.1 region, which encompasses *cadherin 10* (*CDH10*) and *cadherin 9* (*CDH9*), encodes neuronal cell-adhesion molecules and was identified by a genome-wide association approach using 943 ASD families [169]. Although the 2.2 megabase intergenic region between *CDH10* and *CDH9* was not likely to be related to ASD, *CDH10* deletion is one of the common variants that is implicated in ASD risk [159].

Arking et al. identified common variants that contribute to autism in *contactin-associated protein-like 2* (*CNTNAP2*), a member of the neurexin superfamily [162,179,180]. Although rare, alterations in *neurexin1* (*NRXN1*), which is located on chromosome 2p16.3 (which encodes presynaptic cell adhesion molecules) in ASD patients, including subtle sequence variants in the coding region could contribute to the susceptibility of ASD [162,179,180]. 

Deletion at the 17q12 region is a strong candidate for ASD, as well as schizophrenia and RCAD (renal cysts and diabetes syndrome). The loss of one or more of the 15 genes found in the 17q12 region causes ASD and schizophrenia, and the more genes that are deleted, the more brain development and function becomes affected [172]. The chromosomal CNV by 22q11 deletion, which causes 22q11.2 deletion syndromes, shows higher susceptibility for ASD than the general population at about 23%–50% [166]. 

Recently, Leblond et al. found notable ASD-related genes, including *kalirin RhoGEF kinase* (*KALRN*), *phospholipase A2 group IVA* (*PLA2G4A*), and *regulating synaptic membrane exocytosis 4* (*RIMS4*), which are expressed in the nervous system during development and maturation [166]. *KALRN* encodes a guanine nucleotide exchange factor expressed in neural tissue [166,167]. *KALRN*, together with Huntingtin, regulates dendritic spine plasticity, and its de novo variant (3q21.2 duplication) was observed in ASD patients [181]. Both *KALRN* and *PLA2G4A* knockout mouse models showed abnormalities in neuronal maturation and long-term potentiation in the brain [168]. De novo CNV by stop truncation in *RIMS4*, which encodes presynaptic proteins during dendritic and axon morphogenesis, is likely to be associated with perturbed modulation of the releases of glutamate at the synapse and contribute to the development of autism [173]. 

Fragile X mental retardation protein (FMRP), which is associated with Fragile X Syndrome, is a selective RNA binding protein and regulates polyribosome-mediated translation at synapses [182]. Darnell et al. revealed that FMRP interacts with the transcripts involved in ASD [177]. *Fmr1* knockout mice were defective in synaptic plasticity, which may be caused by blocking the translation of proteins with synaptic function [178]. 

Splicing factor nSR100/SRRM4 functions specifically in neuronal cells and regulates neural ‘microexons’ (3–15 nucleotides) through alternative splicing [171]. In ASD patients, neural microexons are frequently misregulated in the brain, which is associated with reduced levels of nSR100. Mirzaa et al. identified 23 variants of *ZNF292* (*zinc finger protein 292*), which resulted from alternative splicing of the most terminal exon 8. These variants in *ZNF292* were associated with neurodevelopmental disorders with or without ASD [170]. Deletion of 6q14.3 in *ZNF292* also resulted in ASD symptoms, such as learning and intellectual disabilities and behavioral problems [183]. 

*Ubiquitin-protein ligase E3A* (*UBE3A*) is one of the most potent regulators involved in ASD pathology. *UBE3A* and neighboring ASD candidate gene *gamma-aminobutyric acid receptor subunit beta-3* (*GABRB3*) were downregulated in *MeCP2*-deficient mice and ASD patients [184]. *UBE3A* is also reduced in Angelman syndrome, which is caused by maternal loss of chromosome 15q11-q13 and shows ASD features [185]. In addition, *UBE3A* is involved in the maintenance of synaptic plasticity and in dendritic spine density [186]. Smith et al. generated a mouse model expressing double and triple doses of *UBE3A*, which was reminiscent of ASD patients with maternal 15q11-13 duplication [187]. Like ASD patients, mice with increased dosage of *UBE3A* showed defective social interaction; they rarely communicated with other mice and did not emit ultrasonic vocalization when they encountered new mice of the same sex.

The deletion of 142 kb at intron 8 of *SHANK3* of the paternal chromosome 22q13 (heterozygous mutation) can cause ASD [95,96,97]. On the other hand, an additional copy of *22q13/SHANK3* did not display ASD symptoms, such as language and social communication impairment [95,97,174,175]. In the postsynaptic density (PSD) complex, *SHANK3* binds to neuroligins (NLGNs) to form glutamatergic synapses [97]. The representative NLGNs, NLGN3 and NLGN4, are essential for cell adhesion and synaptogenesis, and thus, deletion of the NLGN3 gene shows the same phenomenon as autism [176]. Therefore, these reports suggest that *SHANK* together with NLGNs are involved in the formation of the appropriate postsynaptic structure, which is required for the development of language and social communication.

## 4. Epigenetic Dysregulation Underlying ASD

Epigenetic mechanisms regulate chromatin structure and gene expression without altering the DNA sequence [188,189]. They play an important role in the fine-tuning of development-related genes and are involved in the development of the brain; thus, epigenetic dysregulation can cause neurodevelopmental disorders, including ASD. 

Significant differences in expression levels of epigenetic-related genes were found in ASD patients but not in normal individuals, suggesting that epigenetic modifications play a pivotal role in the ASD phenotype [190,191,192]. Only recently, the etiological role of epigenetic dysregulation in ASD has been documented by finding a specific mutation in epigenetic-regulation-related genes in ASD patients. There are two major molecular epigenetic mechanisms involved in gene expression: DNA methylation and histone modification. Noncoding RNA is also a crucial player involved in regulating chromatin structure and gene expression (Table 2).

### 4.1. DNA Methylation

Since DNA methylation links between genes and environmental factors and can answer the complex pathogenesis of autism, most studies have investigated this epigenetic mechanism [206]. Although definitive biological markers or mechanisms underlying ASD have not yet been identified, researchers have been investigating the relationship of environmental exposure and DNA methylation with autism [207,208,209,210,211]. Wong et al. found variant DNA methylation patterns in ASD-discordant monozygotic twins, which is known to be the first epigenetic analysis in ASD patients [212]. They identified differentially methylated CpG sites that were likely to be associated with ASD by comparing ASD-discordant monozygotic twins. Top-ranked differentially methylated regions included *GABRB3*, *AF4/FMR2 family member 2* (*AFF2*), *NLGN2*, *jumonji domain-containing 1C* (JMJD1C), *small nuclear ribonucleoprotein polypeptide N* (SNRPN), *SNRPN upstream reading frame* (SNURF), *UBE3A,* and *potassium inwardly rectifying channel subfamily J member 10* (KCNJ10); some of these were known to be previously implicated in ASD. These findings indicate that DNA methylation as an epigenetic factor can provide an explanation for the etiology of ASD, which may otherwise be difficult to do using a genetic approach. 

DNA methylation at the fifth carbon of the cytosine (5-methylcytosine, 5mC) can be converted to 5-hydroxy methylcytosine (5hmC) during the DNA demethylation process [213]. Interestingly, DNA hydroxymethylation has also been implicated in ASD [214,215]. This was supported by studies that used animal models with mutations in ASD-related genes. Chromatin remodelers *AT-rich interaction domain 1B* (*Arid1b*) and *chromodomain helicase DNA-binding protein 8* (*Chd8*) [216,217,218,219,220,221], histone methyltransferase *euchromatic histone lysine methyltransferase 1* (*Ehmt1*) [222,223], and transcriptional regulators *Foxp1* and *Foxp2* [224,225,226,227,228,229,230,231] were targeted to investigate ASD-like behavioral phenotypes in mutated mice. *SET domain-containing 5* (*Setd5*) was suggested as one of the histone methyltransferase candidates [232], but methyltransferase activity of *Setd5* was not accepted by the majority of other researchers [233,234]. 

Phenotypes of these mutant mice were characterized by sensory disorders, motor coordination disorders, hydrocephalus, and weight loss, as seen in ASD patients. In behavioral tests, they showed increased anxiety-like behavior, social deficits, and repetitive behaviors. In addition, studies have shown that 5hmC is found abundantly in many genes associated with neural development in ASD, including *glutamic acid decarboxylase 67* (*GAD1*) and *RELN* [235]. Of note, DNA methylation at CpG and non-CpG sites has been suggested as a major factor identifying the causes of many other neurological disorders, including Alzheimer’s disease, Parkinson’s disease, Rett syndrome, fragile X syndrome, Huntington’s disease, and amyotrophic lateral sclerosis [211].

#### 4.1.1. MeCP2

One of the best-studied epigenetic factors associated with ASD is *methyl-CpG binding protein 2* (*MeCP2*). *MeCP2* is an important epigenetic regulator of human brain development and is highly abundant in the central nervous system, particularly in GABAergic interneurons. Since the MeCP2 protein has the dual function of acting as both activator and repressor of transcription, the binding action of *MeCP2* in healthy individuals has been shown to regulate many genes with synaptic functions, such as *GABRB3*, *brain-derived neurotrophic factor* (*BDNF*), *distal-less homeobox 5* (*DLX5*), *insulin-like growth-factor-binding protein 3* (*IGFBP3*), *cyclin-dependent kinase-like 1* (*CDKL1*), *protocadherin beta 1* (*PCDHB1*), *protocadherin 7* (*PCDH7*), and *lin-7 homolog A* (*LIN7A*) [184,193]. Abnormal physiological levels of *MeCP2* caused by overexpression via gene duplication or loss of expression by mutation, i.e., *MeCP2* duplication syndrome or Rett syndrome, respectively, were known to exhibit social behavioral disorders similar to ASD [236]. Decreased expression of *MeCP2* in the frontal cortex of ASD patients was associated with abnormal methylation on the promoter of *MeCP2* [237]. This was further supported by Lu et al., demonstrating the important role of *MeCP2* promoter methylation in ASD etiology by locus-specific methylation of the *MeCP2* promotor using dCas9-based methylation targeting method [238]. Kuwano et al. analyzed peripheral blood to determine differentially expressed genes between ASD patients and gender-matched healthy controls. They found that *MeCP2* overexpression (>1.5-fold) was observed in peripheral blood of idiopathic ASD patients [239]. *MeCP2* proteins bind to their target genes and recruit other chromatin remodelers, which may repress the target genes [240]. Zhubi et al. showed that enhanced binding of *MeCP2* to promoters of target genes was correlated with the increased ratio of 5-hydroxymethyl cytosine to 5-methyl cytosine (5hmC/5mC) at the regulatory regions [235]. They also found enhanced *MeCP2* binding to the increased 5hmC/5mC promoter regions of *GAD1* and *RELN* genes, which were known to be downregulated in postmortem brains of ASD patients [235]. *GAD1* is an enzyme that catalyzes glutamate to the inhibitory neurotransmitter γ-aminobutyric acid (GABA) and thereby plays a pivotal role in maintaining an excitatory–inhibitory balance [241,242]. Reelin is a signaling protein that is involved in neural migration, development of neural connection, and modulating synaptic plasticity [243]. 

#### 4.1.2. OXTR

*Oxytocin receptors* (*OXTR*) encode G-protein-coupled receptors that bind to the neurotransmitter peptide hormone oxytocin. *OXTR* is also involved in ASD etiology [194]. Genome-wide microarray and comparative genomic hybridizations on 119 proband of ASD families identified CNV (deletion) of the *OXTR* gene [244]. Interestingly, however, deletion at the *OXTR* region was not detected in the affected sibling of the proband, but aberrant gene silencing by increased DNA methylation at the regulatory region was observed instead. In-depth DNA methylation analysis showed that several CpG islands that regulate *OXTR* expression were hypermethylated in the temporal cortex of ASD patients [244]. This finding was supported by a study that observed hypermethylation at the *OXTR* region in a fetal membrane of preterm birth, which is also an ASD symptom [245]. In addition, adults with ASD showed higher levels of *OXTR* methylation in CpG 16 in the intron 1 region compared to neurotypical subjects. The researchers found that the methylation of CpG 16 was particularly correlated with social interaction and communication scores [246].

#### 4.1.3. SHANK3

Mutations in the *SH3 and multiple ankyrin repeat domains 3* (*SHANK3*) gene are associated with autism and affect the morphology of dendritic spines and synaptic transmission [196]. *SHANK3* is a scaffolding protein in the postsynaptic density and functions in synapse formation and maintenance. Of interest, methylation of CpG island was shown as a strong regulator for *SHANK3* expression. During mouse brain development, *SHANK3* was upregulated two weeks after birth when the methylation rate in the CpG island was highly increased. Zhu et al. reported altered methylation patterns in *SHANK3* by analyzing the DNA methylation profiles of five CpG island regions (CGI-1 to CGI-5) in postmortem brains of ASD patients and controls [247]. Significant increase in overall DNA methylation of CGI at CGI-2, CGI-3, and CGI-4 was found in ASD brain tissues. In addition, the *SHANK3* knockout mouse model exhibits a rescued behavioral phenotype when treated with potent histone deacetylase inhibitor, which strengthens the role of epigenetics in ASD [195]. These reports strongly suggest that defects in DNA methylation and histone modification of ASD-related genes are the underlying mechanisms for the development or symptoms of ASD [195].

### 4.2. Histone Modification and Chromatin Remodeling

Dysregulation of proteins that control histone modifications are associated with ASD. In general, H3K4me3, the trimethylation on the fourth lysine residue of histone H3, plays an important role in the open chromatin formation and gene activation. Specifically, H3K4me3 recruits chromosome remodeling factors to the gene transcriptional start site and is involved in the regulation of the differentiation, growth, and plasticity required for learning and memory of the hippocampus [248,249]. Shulha et al. found that changes in H3K4me3 levels in neurons were related to autism through deep sequencing with anti-H3K4me3-ChIP using prefrontal cortex neurons isolated from postmortem tissue of 6 months to 70 years of age [250]. However, further studies on a larger population are needed to clearly identify and assess the role of H3K4me3 in autism pathophysiology [250]. Similarly, deficiency of *lysine-specific demethylase 5C* (*KDM5C*) alters the epigenetic state, which is associated with intellectual disability and frequent autistic behavior [197]. Duffney et al. also reported that depletion of linker histone H1.4, which is encoded by the *histone cluster 1 H1 family member e* (*HIST1H1E*) gene, is associated with the features of ASD and intellectual disorders [198]. They found a de novo mutation of *HIST1H1E* gene in a 10-year-old boy with ASD [198]. The main function of the H1 linker protein is to organize the higher-order chromatin structure and regulation of gene transcription.

In addition, many efforts have been made to identify autism-related genes using whole exome sequencing in ASD subjects [251]. Frequent mutations of *chromodomain helicase DNA-binding protein* (*CHD*) encoding the ATP-dependent helicase, which is typically involved in chromatin remodeling, occur in autistic individuals [251]. Autistic patients with *CHD8* mutations often showed additional distinct phenotypes, including macrophage and gastrointestinal disorders [252]. *CHD8* appeared to inhibit the target genes of Wnt/β-catenin, with many other CHD8 targets involving autism risk genes [199].

In addition to the CHD family, there are several more genes related to chromatin remodelers, including *ARID1B, BAF chromatin-remodeling complex subunit BCL11A* (*BCL11A*) and *activity-dependent neuroprotector homeobox* (*ADNP*) [201]. *ARID1B*, a component of the ATP-dependent human SWI/SNF (or BAF) chromatin-remodeling complex, is a gene that is frequently mutated in autism [15,200]. In addition, *BCL11A* and *ADNP* are known to encode proteins that interact directly with members of the SWI/SNF complex and have been found to be frequently mutated in autism [201]. Therefore, *ADNP* can be a SWI/SNF-related gene that is ASD-associated and may explain the etiology of about 0.17% of ASD patients [201]. Many other chromatin-remodeling factors, including HDAC4 and polycomb group protein EZH2, are often mutated in patients with intellectual disabilities and ASD, as chromatin regulators are functionally essential for neural progenitor self-renewal, neural differentiation, synaptogenesis, apoptosis, and neurological and cognitive development [253].

### 4.3. MicroRNAs

MicroRNAs (miRNAs) are short noncoding RNA molecules that range from 15 to 22 nucleotides. miRNAs are epigenetic regulators that control the expression of many genes at the level of post-transcription by blocking protein synthesis or inducing mRNA degradation [254]. It is well known that 50% of human genes are regulated by miRNAs and control all the functional pathways involved in cell differentiation, proliferation, development, and apoptosis [202]. To date, about half of all miRNAs identified in humans are expressed in the brain. Mor et al. utilized small RNA sequencing analysis to find unregulated miRNAs and correlated the results with genome-wide DNA methylation data. miRNAs significantly expressed in the ASD brains were associated with synaptic function [255]. Animal model studies also showed that deregulation of miRNA synthesis leads to neurodevelopmental disorders [202,203,204,205]. Abu-Elneel et al. identified 28 miRNAs (out of 466 miRNAs examined) that were differentially expressed between autism and control in a postmortem brain analysis [256]. The differential expression of miRNAs in autistic individuals was also examined in whole blood and lymphoblastoid cell samples [257,258]. 

The treatment of ASD using miRNA-based therapy is a promising strategy because miRNAs can be delivered into cells and not induce integration into the host genome. Overexpressed miRNA in ASD patients could be downregulated by miRNA antagonists, i.e., miRNA inhibition therapy [259], while miRNA replacement therapy using miRNA mimics can compensate for low-expressing miRNAs [259].

## 5. Social Interaction Genes Associated with ASD

As ASD causes defects in social interaction, communication, repetitive patterns of behavior, and lack of attention, it is classified as both a mental disorder and a neurodevelopmental disorder by the American Psychiatric Association. Thus, the prosocial hormone oxytocin has been implicated with the pathogenesis and treatment of ASD. Oxytocin and vasopressin systems were proposed as modulators of social behavior in vertebrates, including humans [260]. Significantly higher oxytocin and receptor binding was observed in the nucleus basalis of Meynert of the forebrain of ASD frozen specimens [261]. On the other hand, significantly lower binding of oxytocin to receptor was detected in the ventral pallidum (VP) of ASD brains compared to controls. VP is involved in the mesolimbic dopamine reward pathway, and lower oxytocin receptor binding levels may be interpreted as a reduced experience of oxytocin-mediated social reward in ASD patients [261]. Of note, ASD patients who inhaled a single dose of oxytocin displayed promoted social behavior [262].

Variation in the *arginine vasopressin receptor 1a* (*AVPR1a*) gene on chromosome 12q14–15, which encodes the V1a receptor, is also associated with a deficiency in social behavior. As stated above, vasopressin, a neuropeptide, has been implicated in the social adaptation of mammals, including humans [263]. In terms of ASD diagnosis, *AVPR1a* variation is correlated with the scores of autism quotients [264]. Independent reports suggest that microsatellites in the promoter region of *AVPR1a* are closely related to ASD [265]. Intravenous administration of RG7713, a V1a receptor antagonist, could improve deficiencies in social communication in ASD without intellectual disability [266]. 

## 6. Conclusion and Future Perspectives

ASD is a complex neurodevelopmental disorder with diverse symptoms and various aspects. Therefore, it is likely that various factors are involved in ASD, including environmental, genetic, and epigenetic. Presently, a genomic approach using patient samples may be the most appropriate method as ASD displayed the highest proportion of cases with a clinically relevant CNV [267]. However, since epigenetics is the underlying mechanism of regulating gene expression, it will be necessary to understand ASD from an epigenetic perspective, which will help to develop ASD therapies by controlling epigenetic states.

The human brain is a complex organ composed of various types of neurons, glia, microglia, neuroepithelial cells, neural stem/progenitor cells, etc., consisting of about 100 billion cells. Moreover, since human brains grow and change in proportion and composition throughout life, particularly early in life, it may be difficult to understand the exact etiology of ASD by studying postmortem samples. Thus, induced pluripotent stem cells (iPSCs) may be an alternative way to study the human brain or neurological disorders [268]. As iPSCs are pluripotent and can differentiate into all cell types of our body, patient-derived iPSCs can be used as starting material to generate disease model neurons or three-dimensional minibrain structures, i.e., brain organoids [269]. For example, Mariani et al. utilized brain organoids to identify the mechanism underlying ASD [270]. Brain organoids generated using ASD-patient-derived iPSCs showed accelerated cell cycles and an increased number of GABAergic neurons, which was controlled by *forkhead box G1* (*FOXG1*). Therefore, *FOXG1* could be a candidate gene for the etiology underlying ASD.

ASD manifests complex phenotypes and is usually accompanied by comorbidities. One of the comorbidities associated with ASD is gastrointestinal problems, which are closely affected by the gut microbiome [271]. Recent studies suggest that intestinal microbiome is involved in many neurological disorders, including Alzheimer’s disease and ASD, and have built upon the concept of the brain–gut axis, in which the brain and gastrointestinal tract communicate bidirectionally via signaling molecules [272]. Although much remains to be discovered about the brain–gut-microbiome axis, manipulating the gut microbiota composition in patients with ASD may be a potential therapy for treating gastrointestinal problems in ASD [271].

Another intriguing approach to ameliorate ASD symptoms is to use exosomes, which are suborganelles rich in DNA, RNA, and protein content [273]. Interestingly, the composition of exosomes secreted from mesenchymal stem cells could be manipulated by the treatment of several interleukins [274]. Recently, neuroinflammation induced by proinflammatory cytokines was suggested as a novel pathogenesis of ASD, particularly in irritability and socialization problems [275]. As a therapeutic approach, stem-cell-derived exosomes containing anti-inflammatory molecules could be used as efficient carriers for delivering anti-inflammatory molecules across the blood–brain barrier [276]. These attempts have already been made to alleviate the symptoms of Alzheimer’s disease [276]. 

Recent developments in bioengineering and stem cell technology will help to understand the pathophysiology and cure of neurological diseases, including ASD. Although much effort has been made to identify the pathogenesis of ASD, research using organoids is expected to be responsible for the future of medical science advancements. The combinatorial approach, by which several different tissue organoids from patient-derived iPSCs are cultured in microfluidic organ-on-a-chip system, has great potential in allowing the study of organ physiology and disease etiology in a simulated tissue–tissue interaction system [277].

## Figures and Tables

**Figure 1 jcm-09-00966-f001:**
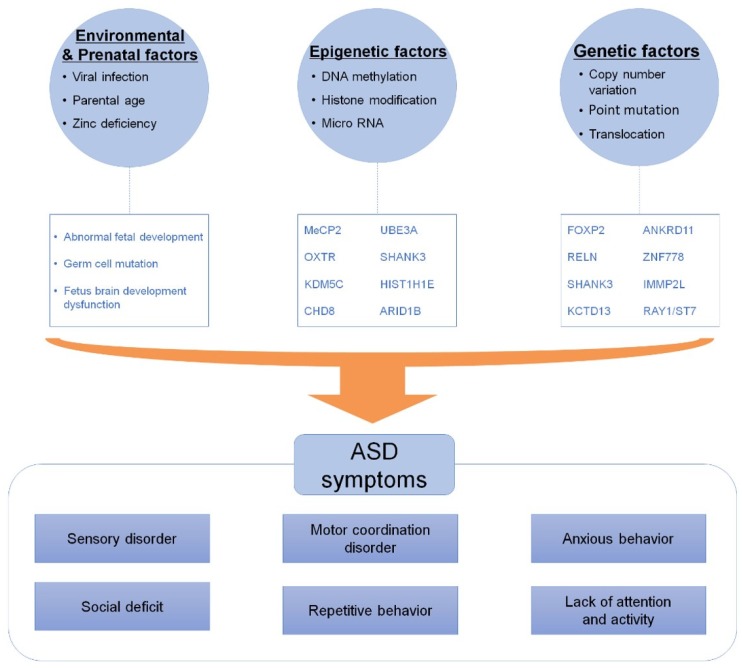
Comprehensive overview of the diverse etiology of autism spectrum disorder (ASD). Although definitive etiology and pathogenesis underlying ASD have not yet been identified, accumulated evidence has identified various risk factors, including environmental, genetic, and epigenetic factors.

**Figure 2 jcm-09-00966-f002:**
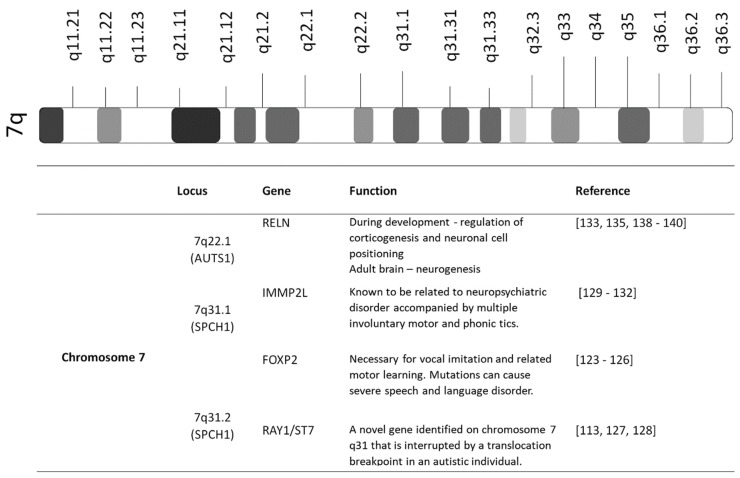
Loci on chromosome 7 responsible for autism spectrum disorder (ASD). Chromosome 7 contains more ASD-related regions than other chromosomes. Genome-wide association studies found SPCH1 and AUTS1 (autism susceptibility locus), which encompass *RELN*, *IMMP2L*, *FOXP2*, and *RAY1/ST7*, to be the specific loci responsible for the defect in speech and language development in ASD patients.

**Table 1 jcm-09-00966-t001:** Chromosome locus associated with ASD.

Locus	Function	Genes	Variation	Reference
3q21.1 - 3q21.2	Abnormalities in neuronal maturation and long-term potentiation in the brain, macrocephaly, intellectual disability facial dysmorphism	KALRN	Duplication	[166,167,168]
5p14.1	Neuronal cell-adhesion molecules	Cadherin 10 (CDH10) Cadherin 9 (CDH9)	Deletion	[159,169]
6q14.3	Learning problems, intellectual disability, behavioral problems	ZNF292 (zinc finger protein 292)	Deletion	[170]
12q24.23	Neuronal cells and misregulated neural ‘microexons’ in the brains	nSR100/SRRM4		[171]
16p11.2	Reduced proliferation of neuronal progenitors, the increased cell death during the brain development, microcephaly	KCTD13	Deletion Duplication	[161,162,163,164,165]
16q24.3	Cognitive impairment, brain abnormality	ANKRD11 ZNF778	Microdeletion	[160]
17q12	Macrocephaly, neurocognitive impairment	HNF1B	Deletion	[172]
20q13.12	Releases of glutamate at the synapse	RIMS4		[166,173]
22q11.2	Physical, behavioral, social communication, neurocognitive impairments		Deletion	[166]
22q13	Cognitive deficits, behavioral autistic symptoms, language and social communication problems	SHANK3	Deletion	[96,97,174,175,176]
Xq27.3	Synaptic function in the brain	FMR1		[166,177,178]

**Table 2 jcm-09-00966-t002:** Epigenetic factors implicated in ASD.

Epigenetic Factors	Genes	Function	Possible Epigenetic Mechanisms	Reference
DNA methylation	MeCP2	Encodes a methyl binding protein that binds to the methylated region of DNA and silence the gene. Has a role in synaptic development and long-term synaptic plasticity.	MeCP2 regulation of other genes via epigenetics: recruitment of co-repressors, chromatin looping.	[184,193]
	UBE3A	Known for its role in Angelman syndrome.	Loss of imprinting of one copy, and production of antisense RNA that binds to UBE3A and mRNA Prevents translation.	[185,186]
	OXTR	G-protein coupled receptor for oxytocin. Modulates: stress, anxiety, social memory, maternal-offspring behavior, etc.	Hypermethylation and silencing.Decreased OXTR expression.	[194,195]
	SHANK3	Effect on the morphology of dendritic spine and synaptic transmission	Expression of SHANK3 was strongly regulated by methylated CpG island.	[195,196]
Histone modification	KDM5C	Alters the epigenetic state, which is associated with intellectual disability and frequent autistic behavior.	Involved in the regulation of transcription and chromatin remodeling.	[197]
	HIST1H1E	Associated with the features of ASD and intellectual disorders.	To organize the higher-order chromatin structure and regulation of gene transcription.	[198]
	CHD8	Inhibit the target genes of Wnt/β-catenin, and many of the genes in CHD8 targets included autism risk genes.	Encode ATP-dependent helicases that are typically involved in chromatin remodeling.	[199]
	ARID1B	A component of the ATP-dependent human SWI/SNF chromatin-remodeling complex.	Involved in chromatin remodeling.	[15,200]
	BCL11A	Encode proteins that interact directly with members of the SWI/SNF.	Involved in chromatin remodeling.	[201]
	ADNP	Encode proteins that interact directly with members of the SWI/SNF.	Involved in chromatin remodeling.	[201]
Micro RNA		Deregulation of miRNA synthesis leads to neurodevelopmental disorders.	Epigenetic regulator that control the expression of many genes at the level of post-transcription by blocking protein synthesis or mRNA degradation.	[202,203,204,205]

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
