# Peer review of "Genetic and Epigenetic Etiology Underlying Autism Spectrum Disorder"

_jcm, 2020, doi:10.3390/jcm9040966_

Round 1

Reviewer 1 Report

Comments

The authors have made a review about the environmental, genetic and epigenetic factors that have been involved with the cause of Autism Spectrum Disorder (ASD). I consider that the review is well conducted in general. However, there are some points that should be modified.

The first and more important point is the paragraph about the link between ASD and vaccination (sentences 72-78). We, as scientists, should be very careful on this topic, as it is not based in valid scientific reports. The fact that the authors even cite twice the retracted paper that originated this alarm in the society (and caused that parents refuse to vaccinate their kids based on bad scientific reports, so they prefer that their kids get measles), should be seriously reconsidered. A retracted paper should be consider as not published, and hence, not citable.

Apart of that, there are some sentences that should be moved in the text, as they are not logical where they are. For instance, the sentence 73-75 should be transferred to the previous paragraph; sentence 384-386 should be moved to the end of the paragraph.

It is also frequent in the paper to find acronyms that have not been spelled out before, such as CNV, gene names, ChIP... Please correct this.

Concerning the table 1, the name of gene for FMRP (this is the name of the protein) should be changed to FMR1. This dichotomy should be also explained in the main text. Moreover, what is the rationale behind the order of the loci in this table? Relevance? If so, indicate it in the main text. As it is, it looks more like a random list of genomic locations.

Please, check the correct spelling of the genes. For instance, line 163: “RENL” should be “RELN”; line 414: “ARTP1B” should be “ARID1B”.

Additional comments:

Lines 92-94: check sentence

Line 109: “… it was expected that Zinc could cause damage to the nervous system”. Do you mean “… it was expected that an excess of Zinc could cause damage to the nervous system”?

Line 180: check sentence

Line 266: What do the authors mean with “…regulates polyribosome association in the brain”? “polysome mediated translation”?

Lines 269-270: check sentence

Line 288: Do the authors mean “…such as language and social communication impairment”?

Lines 356-358: check sentence

Line 291: “…fourth lysine residue of histone H3

Line 398: check sentence

Lines 407-409: check sentence

Lines 419-423: check sentence

Line 427: “…by blocking protein synthesis or inducing mRNA degradation”

Line 462-463: Is this sentence required?

Lines 482-483: this sentence should be down tuned

Table 1: “q21.2” should be “3q21.2”

Table 2: “NA methylation” should be “DNA methylation”

Author Response

Comments

The authors have made a review about the environmental, genetic and epigenetic factors that have been involved with the cause of Autism Spectrum Disorder (ASD). I consider that the review is well conducted in general. However, there are some points that should be modified.

The first and more important point is the paragraph about the link between ASD and vaccination (sentences 72-78). We, as scientists, should be very careful on this topic, as it is not based in valid scientific reports. The fact that the authors even cite twice the retracted paper that originated this alarm in the society (and caused that parents refuse to vaccinate their kids based on bad scientific reports, so they prefer that their kids get measles), should be seriously reconsidered. A retracted paper should be consider as not published, and hence, not citable.

â–¶ Yes, we agree with your comment. We have deleted the retracted paper and rephrased the sentence.

Apart of that, there are some sentences that should be moved in the text, as they are not logical where they are. For instance, the sentence 73-75 should be transferred to the previous paragraph; sentence 384-386 should be moved to the end of the paragraph.

â–¶ We have moved the sentences to where you suggested.

It is also frequent in the paper to find acronyms that have not been spelled out before, such as CNV, gene names, ChIP... Please correct this.

â–¶ The acronyms have been all spelled out in revision.

Concerning the table 1, the name of gene for FMRP (this is the name of the protein) should be changed to FMR1. This dichotomy should be also explained in the main text. Moreover, what is the rationale behind the order of the loci in this table? Relevance? If so, indicate it in the main text. As it is, it looks more like a random list of genomic locations.

â–¶ We have corrected FMRP to FMR1. Tables 1 has been rearranged in a numerical order.

Please, check the correct spelling of the genes. For instance, line 163: “RENL” should be “RELN”; line 414: “ARTP1B” should be “ARID1B”.

â–¶ We have checked and corrected the typos.

Additional comments:

Lines 92-94: check sentence

â–¶ The sentence has been checked and cited with appropriate references.

Line 109: “… it was expected that Zinc could cause damage to the nervous system”. Do you mean “… it was expected that an excess of Zinc could cause damage to the nervous system”?

â–¶ Yes, that means "an excess of Zinc". We have corrected this point.

Line 180: check sentence

â–¶ The sentence has been cited with appropriate references.

Line 266: What do the authors mean with “…regulates polyribosome association in the brain”? “polysome mediated translation”?

â–¶ The sentence has been changed as follows:

Fragile X mental retardation protein (FMRP), which is associated with Fragile X Syndrome, is an selective RNA binding protein and regulate polyribosome mediated translation at synapses

Lines 269-270: check sentence

â–¶ We have checked and corrected the typos.

Line 288: Do the authors mean “…such as language and social communication impairment”?

â–¶ Yes, it is " such as language and social communication impairment". We have corrected this.

Lines 356-358: check sentence

â–¶ The sentence has been changed as follows:

Zhubi et al. showed that enhanced binding of MeCP2 to promoters of target genes was correlated with the increased ratio of 5-hybroxymethyl cytosine to 5-methyl cytosine (5hmC/5mC) at the regulatory regions[213].

Line 391: “…fourth lysine residue of histone H3

â–¶ We have checked and corrected the typo.

Line 398: check sentence

â–¶ We have checked the sentence, but we could not see any grammatic problem. We have just cited with appropriate references.

Lines 407-409: check sentence

â–¶ The sentence has been changed as follows:

Frequent mutations of Chromodomain helicase DNA-binding protein (CHD) encoding the ATP-dependent helicase, which is typically involved in chromatin remodeling, occur in autistic individuals[233].

Lines 419-423: check sentence

â–¶ The sentence has been changed as follows:

Many other chromatin remodeling factors, including HDAC4 and Polycomb group protein EZH2, are often mutated in patients with intellectual disabilities and ASD, as chromatin regulators are functionally essential for neural progenitor self-renewal, neural differentiation, synaptogenesis, apoptosis, and neurological and cognitive development[238].

Line 427: “…by blocking protein synthesis or inducing mRNA degradation”

â–¶ We have corrected the sentence according to the reviewer's comment.

Line 462-463: Is this sentence required?

â–¶ We have removed this sentence.

Lines 482-483: this sentence should be down tuned

â–¶ The sentence has been down tuned as follows:

Brain organoids generated using ASD patient-derived iPSCs showed accelerated cell cycles and an increased number of GABAergic neurons, which was controlled by FOXG1. Therefore, FOXG1 could be a candidate gene for the etiology underlying ASD.

Table 1: “q21.2” should be “3q21.2”

â–¶ We have checked and corrected the typo.

Table 2: “NA methylation” should be “DNA methylation”

â–¶ We have checked and corrected the typo.

Reviewer 2 Report

The manuscript by Yoon and colleagues provides a detailed description of some environmental and genetic risk factors for autism. However, this reviewer feels like this review is not completely up-to-date with the current literature. Only 20% of the citations of this manuscript refers to papers that were published in the last five years. Also thanks to the increasing usage of next generations sequencing technologies, significant advancements have been made in the past five years in the comprehension of the genetic causes of ASD. Accordingly, the Simons Foundation Autism Research Initiative (SFARI) database currently reports 913 genes implicated with autism and 17 recurrent CNV loci. In this context, the 7q22-q33 region described by the authors as a chromosomal locus that affects ASD is currently not considered a recurrent CNV locus in individuals with autism. Among the genes presented by the authors as “the most plausible candidate genes for the causation of autism” (page 4, row 141), only RELN seems to have a strong association with ASD and has received a SFARI score equal to 1. For the other three genes, there are also studies that describe a negative association between these genes and ASD (https://www.ncbi.nlm.nih.gov/pubmed/17043892; https://www.ncbi.nlm.nih.gov/pubmed/24599690; https://www.ncbi.nlm.nih.gov/pubmed/11894222; https://www.ncbi.nlm.nih.gov/pubmed/23277129).

In addition, whereas paternal age and maternal infections during pregnancy seem to be widely accepted risk factors for ASD, the role of zinc deficiency is still controversial (https://www.ncbi.nlm.nih.gov/pubmed/30821006; https://www.ncbi.nlm.nih.gov/pubmed/28883881). For these reasons, the authors should update their manuscript with more recent data.

Furthermore, some conceptual imprecision are also present and need to be corrected:

  • Page 7, row 212: “in addition to genetic cause, recent studies have found that inherited and de novo CNV could contribute to ASD”. CNV are by all means a genetic cause of ASD
  • Page 9, row 300: “significant overexpression of epigenetic-related genes were found in ASD patients but not in normal individuals, suggesting that epigenetic modification play a pivotal role in the ASD phenotype”. To my knowledge, the pathogenic mechanism associated with most of the ASD-causing genes is linked to haploinsufficiency rather than overexpression.
  • Page 11, row 329: “histone methyl transferases Ehmt1 and Setd5”. The role of SETD5 as a methyltransferase is controversial. Currently, only one paper reports a possible methyltransferase activity for SETD5 (https://www.ncbi.nlm.nih.gov/pubmed/31515109). However, the majority of papers concluded that SETD5 does not have methyltransferase activity (https://www.ncbi.nlm.nih.gov/pubmed/30455454; https://www.ncbi.nlm.nih.gov/pubmed/27864380).

Lastly, I would recommend that the authors have someone with English as the primary language to review the manuscript.

Author Response

Comments and Suggestions for Authors

The manuscript by Yoon and colleagues provides a detailed description of some environmental and genetic risk factors for autism. However, this reviewer feels like this review is not completely up-to-date with the current literature. Only 20% of the citations of this manuscript refers to papers that were published in the last five years. Also thanks to the increasing usage of next generations sequencing technologies, significant advancements have been made in the past five years in the comprehension of the genetic causes of ASD. Accordingly, the Simons Foundation Autism Research Initiative (SFARI) database currently reports 913 genes implicated with autism and 17 recurrent CNV loci. In this context, the 7q22-q33 region described by the authors as a chromosomal locus that affects ASD is currently not considered a recurrent CNV locus in individuals with autism. Among the genes presented by the authors as “the most plausible candidate genes for the causation of autism” (page 4, row 141), only RELN seems to have a strong association with ASD and has received a SFARI score equal to 1. For the other three genes, there are also studies that describe a negative association between these genes and ASD (https://www.ncbi.nlm.nih.gov/pubmed/17043892; https://www.ncbi.nlm.nih.gov/pubmed/24599690; https://www.ncbi.nlm.nih.gov/pubmed/11894222; https://www.ncbi.nlm.nih.gov/pubmed/23277129).

â–¶ Each gene mentioned was additionally described using recent papers that were written in the last 5-7 years. As for genes that lacking plausibility for their effects on autism, the reasons for the lack of credibility were further described using appropriate references including those provided by reviewer.

In addition, whereas paternal age and maternal infections during pregnancy seem to be widely accepted risk factors for ASD, the role of zinc deficiency is still controversial (https://www.ncbi.nlm.nih.gov/pubmed/30821006; https://www.ncbi.nlm.nih.gov/pubmed/28883881). For these reasons, the authors should update their manuscript with more recent data.

â–¶ We have added the controversial issue with citation of recent papers according to the reviewer's suggestions.

Furthermore, some conceptual imprecision are also present and need to be corrected:

Page 7, row 212: “in addition to genetic cause, recent studies have found that inherited and de novo CNV could contribute to ASD”. CNV are by all means a genetic cause of ASD

â–¶ Yes, it is redundant. We have removed “genetic cause”.

Page 9, row 300: “significant overexpression of epigenetic-related genes were found in ASD patients but not in normal individuals, suggesting that epigenetic modification play a pivotal role in the ASD phenotype”. To my knowledge, the pathogenic mechanism associated with most of the ASD-causing genes is linked to haploinsufficiency rather than overexpression.

â–¶ Yes, I agree that "overexpression" can be interpreted in a narrow sense. We have changed “overexpression of epigenetic-” to “differentially expressed epigenetic-”.

Page 11, row 329: “histone methyl transferases Ehmt1 and Setd5”. The role of SETD5 as a methyltransferase is controversial. Currently, only one paper reports a possible methyltransferase activity for SETD5 (https://www.ncbi.nlm.nih.gov/pubmed/31515109). However, the majority of papers concluded that SETD5 does not have methyltransferase activity

(https://www.ncbi.nlm.nih.gov/pubmed/30455454; https://www.ncbi.nlm.nih.gov/pubmed/27864380).

â–¶ The sentence has been down tuned as follows:

.. histone methyl transferases euchromatic histone lysine methyltransferase 1 (Ehmt1)[228,229], and transcriptional regulators Foxp1 and Foxp2[230-237] were targeted to investigate ASD-like behavioral phenotypes in mutated mice. SET domain containing 5 (Setd5) was suggested as one of the histone methyl transferase candidates[238], but methyltransferase activity of Setd5 was not accepted by majority of other researches[239,240].

Lastly, I would recommend that the authors have someone with English as the primary language to review the manuscript.

â–¶ We already got English Editing service from Editage (www.editage.co.kr). If there are still many problems with the English expression, we will request a service once again before publication.

Round 2

Reviewer 2 Report

Yoon and colleagues have now introduced some more recent references in their review.

I think that any reference to the correlation between autism and vaccination should be completely removed, as it might result misleading.

I would advise the authors to re-check their manuscript for typos/grammar/style. Here are some examples:

  • Page 1, row 30: “ASD has increased steadily since the term was coined and has increased significantly in the past 20 years”. This sentence is redundant
  • Figure 1, genetic factor figure: the authors may want to refer point mutations instead of “mutation”
  • Page 6, row 196: “provid” should become provide
  • Page 6, row 201: “IMP2 inner mitochondrial membrane protease-like (IMMP2L) was identified as the most associated gene to autism through high-density SNP”. I guess the authors mean “most frequently associated”
  • Page 9, row 292: “is an selective” should be “is a selective”
  • Page 10, row 328: “Significant overexpression differentially expressed of epigenetic-related genes were found in ASD patients but not in normal individuals”. In its current state, this sentence does not make sense. I understand what the authors are trying to say, but the sentence should be re-phrased
  • Page 11, row 341: “methylation works”. It would be more appropriate to say “DNA methylation depends”
  • Page 16, row 522: “One of the comorbidities associated with ASD is gastrointestinal problems”. Here the verb should be are instead of is

Author Response

Yoon and colleagues have now introduced some more recent references in their review.

I think that any reference to the correlation between autism and vaccination should be completely removed, as it might result misleading.

â–¶ We have completely removed the sentence concerning the correlation between autism and vaccination.

On the other hand, many studies have shown that there is no correlation of MMR vaccines to autism although argued by one report. In addition, statistical studies of pregnant women suggest that there is no correlation between maternal vaccination and ASD.

I would advise the authors to re-check their manuscript for typos/grammar/style. Here are some examples:

Page 1, row 30: “ASD has increased steadily since the term was coined and has increased significantly in the past 20 years”. This sentence is redundant

â–¶ We have removed "..and has increased significantly in the past 20 years",.

Figure 1, genetic factor figure: the authors may want to refer point mutations instead of “mutation”

â–¶ We have changed "mutation" to "point mutation"

Page 6, row 196: “provid” should become provide

â–¶ The typo mentioned has been fixed accordingly.

Page 6, row 201: “IMP2 inner mitochondrial membrane protease-like (IMMP2L) was identified as the most associated gene to autism through high-density SNP”. I guess the authors mean “most frequently associated”

â–¶ We have changed "associated" to "frequently associated".

Page 9, row 292: “is an selective” should be “is a selective”

â–¶ The typo has been corrected.

Page 10, row 328: “Significant overexpression differentially expressed of epigenetic-related genes were found in ASD patients but not in normal individuals”. In its current state, this sentence does not make sense. I understand what the authors are trying to say, but the sentence should be re-phrased

â–¶ The sentence has been changed as follows:

Significant differences in expression levels of epigenetic-related genes were found in ASD patients but not in normal individuals,

Page 11, row 341: “methylation works”. It would be more appropriate to say “DNA methylation depends”

â–¶ We have changed "works" to "links".

Page 16, row 522: “One of the comorbidities associated with ASD is gastrointestinal problems”. Here the verb should be are instead of is

â–¶"is" is grammatically correct.